# Deciphering the Mechanism of Silver Catalysis of "Click" Chemistry in Water by Combining Experimental and MEDT Studies †

**Hicham Ben El Ayouchia** [1,*], **Lahoucine Bahsis** [1,2], **Ismail Fichtali** [3], **Luis R. Domingo** [4,*], **Mar Ríos-Gutiérrez** [4], **Miguel Julve** [5] and **Salah-Eddine Stiriba** [1,5,*]

[1]  Laboratoire de Chimie Analytique et Moléculaire, LCAM, Faculté Polydisciplinaire de Safi, Université Cadi Ayyad, Safi 46030, Morocco; bahsis.lahoucine@gmail.com

[2]  Département de Chimie, Faculté des Sciences d'El Jadida, Université Chouaïb Doukkali, El Jadida 24000, Morocco

[3]  Laboratoire de Chimie Organique Appliquée, Université Sidi Mohammed Ben Abdallah, Faculté des Sciences et Techniques, Route Immouzer BP 2202 Fès, Morocco; biocmb@gmail.com

[4]  Departamento de Química Orgánica, Universidad de Valencia, Avda. Dr. Moliner 50, 46100 Burjassot, Valencia, Spain; m.mar.rios@uv.es

[5]  Instituto de Ciencia Molecular/ICMol, Universidad de Valencia, C/Catedrático José Beltrán 2, 46980 Paterna, Valencia, Spain; miguel.julve@uv.es

\*  Correspondence: belayou@gmail.com (H.B.E.A.); domingo@utopia.uv.es (L.R.D.); stiriba@uv.es (S.-E.S.); Tel.: +34-96-354-4445 (S.-E.S.)

†  Dedicated by one of us S.-E.S. to his beloved mom Aicha Lakani, who passed away on 24 October 2019.

**Abstract:** A combined experimental study and molecular electron density theory (MEDT) analysis was carried out to investigate the click of 1,2,3-triazole derivatives by Ag(I)-catalyzed azide-alkyne cycloaddition (AgAAC) reaction as well as its corresponding mechanistic pathway. Such a synthetic protocol leads to the regioselective formation of 1,4-disubstituted-1,2,3-triazoles in the presence of AgCl as catalyst and water as reaction solvent at room temperature and pressure. The MEDT was performed by applying Density Functional Theory (DFT) calculations at both B3LYP/6-31G(d,p) (LANL2DZ for Ag) and ωB97XD/6-311G(d,p) (LANL2DZ for Ag) levels with a view to decipher the observed regioselectivity in AgAAC reactions, and so to set out the number of silver(I) species and their roles in the formation of 1,4-disubstituted-1,2,3-triazoles. The comparison of the values of the energy barriers for the mono- and dinuclear Ag(I)-acetylide in the AgAAC reaction paths shows that the calculated energy barriers of dinuclear processes are smaller than those of the mononuclear one. The type of intramolecular interactions in the investigated AgAAC click chemistry reaction accounts for the regioselective formation of the 1,4-regioisomeric triazole isomer. The ionic character of the starting compounds, namely Ag-acetylide, is revealed for the first time. This finding rules out any type of covalent interaction, involving the silver(I) complexes, along the reaction pathway. Electron localization function (ELF) topological analysis of the electronic structure of the stationary points reaffirmed the *zw*-type (zwitterionic-type) mechanism of the AgAAC reactions.

**Keywords:** click chemistry; AgAAC; 1,2,3-triazole; water; MEDT; ELF; mechanism

---

## 1. Introduction

1,2,3-Triazole-rich molecules have found immense applications in various fields such as medicine, biology, and materials science and engineering [1–6]. In general, these important five-membered heterocycle scaffolds are synthesized by means of azide-alkyne cycloaddition (AAC) reactions [7,8].

In fact, the uncatalyzed [3 + 2] cycloaddition (32CA) reaction, known as "Huisgen's 1,3-dipolar cycloaddition," between azides and alkynes is characterized by a lower rate and moderate yields at room temperature because of the high kinetic barrier of the reaction [9,10]. In 2002, Sharpless and Meldal introduced the copper(I) catalytic species [generated in-situ by the reduction of copper(II)] for the "click" of 1,2,3-triazoles, which results in a major improvement in both rate and regioselectivity of such a cycloaddition process [11,12]. The copper-catalyzed azide-alkyne cycloaddition (CuAAC) reaction was an important advance in the triazole chemistry because of its high regioselectivity, affording only 1,4-disubstituted-1,2,3-triazoles, wide substrate scope and mild reaction conditions [13]. However, the use of copper(I), known for its high cytotoxicity as well as its oxidative damage, decreases its synthetic applicability [14,15]. In 2005, Zang et al. developed a new catalyst based on the [Cp*RuCl(PPh$_3$)$_2$] complex [Cp* = pentamethylcyclopentadienyl and PPh$_3$ = triphenylphosphine] for the preparation of 1,5-disubstituted-1,2,3-triazoles through RuAAC reactions in a regioselective manner [16]. Later, the use of [RuH$_2$(CO)PPh$_3$] enabled the switch of the regiochemical outcome, affording 1,4-disubstituted-1,2,3-triazoles in a regioselective manner [17]. It is noteworthy that the RuAAC reaction requires complex reaction conditions that decrease the synthetic applicability too, making it a relatively expensive method. Likewise, Veige's group reported the click reaction between Au(I)-azides and Au(I)-acetylides resulting in digold triazolates [18,19]. Indeed, the isolation of digold triazolate complexes supported the role of two copper(I) ions in the CuAAC, in a smart manner. The Zn-mediated 32CA reaction of azides and alkynes (ZnAAC) affording regioselective formation of 1,5-triazoles at room temperature was also reported [20,21]. Furthermore, the Ir-catalyzed intermolecular AAC (IrAAC) reaction has also proved to be a valuable complement to the well-known CuAAC and RuAAC reactions, generating regioselective 1,4,5-trisubstituted-1,2,3-triazoles [22].

McNulty et al. published the first example of Ag(I)-catalyzed AAC reactions (hereafter named AgAAC) as an alternative of the CuAAC reactions to regioselectively envisaging the preparation of 1,4-disusbtituted-1,2,3-triazoles at room temperature [23]. Later, this group developed new Ag(I) complexes with lower toxicity, by using Ag(I) with *P,O*-containing ligands [24]. Recently, Cuevas et al. successfully reported AgAAC reactions catalyzed by silver(I) chloride, which were improved by introducing *N*-heterocyclic carbenes [25]. In further experiments, Ag$_2$O nanoparticles as well as the AgN(CN)$_2$/DIPEA system (DIPEA = *N,N*-Diisopropylethylamine) were successfully employed in AgAAC reactions, using water/ethylene glycol mixtures as the reaction medium [26,27]. However, in contrast to the well-studied mechanisms of copper- and ruthenium-catalyzed AAC reactions [28–31], the mechanism of the AgAAC process and its regiochemical outcomes remain unclear and they need deep and systematic research efforts. In this context, Tuzun et al. reported the first theoretical study about the mechanism of the AgAAC reaction based on the work by McNulty's group [32]. This mechanistic study was initiated by analyzing a Ag(I)-acetylide structure (Figure 1); however, the key catalytic intermediates have not been experimentally isolated. Furthermore, the results of the comparison between the pathways that involve mononuclear or dinuclear intermediates showed that the barrier for the dinuclear case is lower than that of the mononuclear one in gas phase, which is similar to the CuAAC reaction analogue that favors the dinuclear pathway [32].

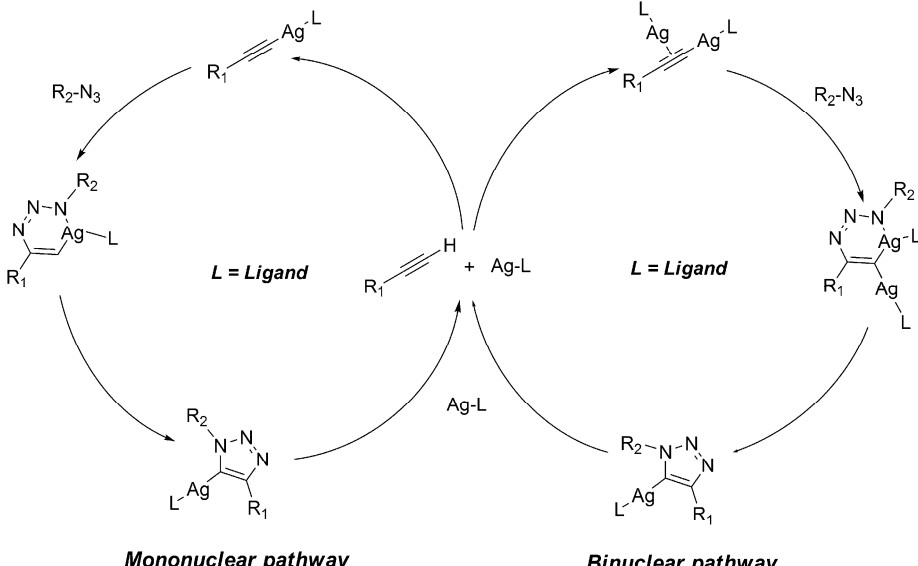

**Figure 1.** Proposed catalytic cycles for the AgAAC reaction.

Recently, we reported a combined theoretical and experimental investigation on the preparation of 1,4-disubstituted-1,2,3-triazoles via CuAAC reactions [33] by Molecular Electron Density Theory (MEDT) [34]. The quest for an alternative metal to copper in AAC that may be both effective and biocompatible led us to investigate the AgAAC reaction by using AgCl as precursor for the silver catalytic species. Silver(I) ions are known for their biocompatibility in biological and medical issues and their anti-microbial properties are well-established. To carry out AgAAC under strict "click" conditions, water was employed as the solvent. Indeed, the presence of silver particles or ions in the obtained triazole derivatives under AgAAC is expected to present no biological inconvenience, in contrast to the presence of copper traces produced under CuAAC. The mechanism and the origin of the regioselectivity of the AgAAC reactions have been systematically addressed through MEDT by using DFT methods.

## 2. Results and Discussion

### 2.1. Setup and Scope of the Click AgAAC

The most promising and successful silver catalyst tested in the 32CA reactions of organic azides with terminal alkynes was found to be AgCl, considering the recent results reported by Banerji et al. [35]. The reaction of benzyl azide (**1a**) and phenylacetylene (**2a**) was initially chosen as a reaction model of our study (Scheme 1). Surprisingly, the result for this 32CA reaction proved that the 1,4-isomer 1,2,3-triazole (**3a**) was formed in high yield (92%) within 24 h with 10 mol% of the AgCl catalyst in the dark under ambient conditions, using water as solvent. A lower yield of **3a** was obtained for the same 32CA reaction when performed under the same conditions, in the presence of light (66% yield), while no 1,2,3-triazole product was obtained in the lack of AgCl in water at room temperature. In addition, the 1,5-isomer 1,2,3-triazole was not detected as confirmed by $^1$H and $^{13}$C NMR analyses of the final crudes of the AgAAC click chemistry reaction type.

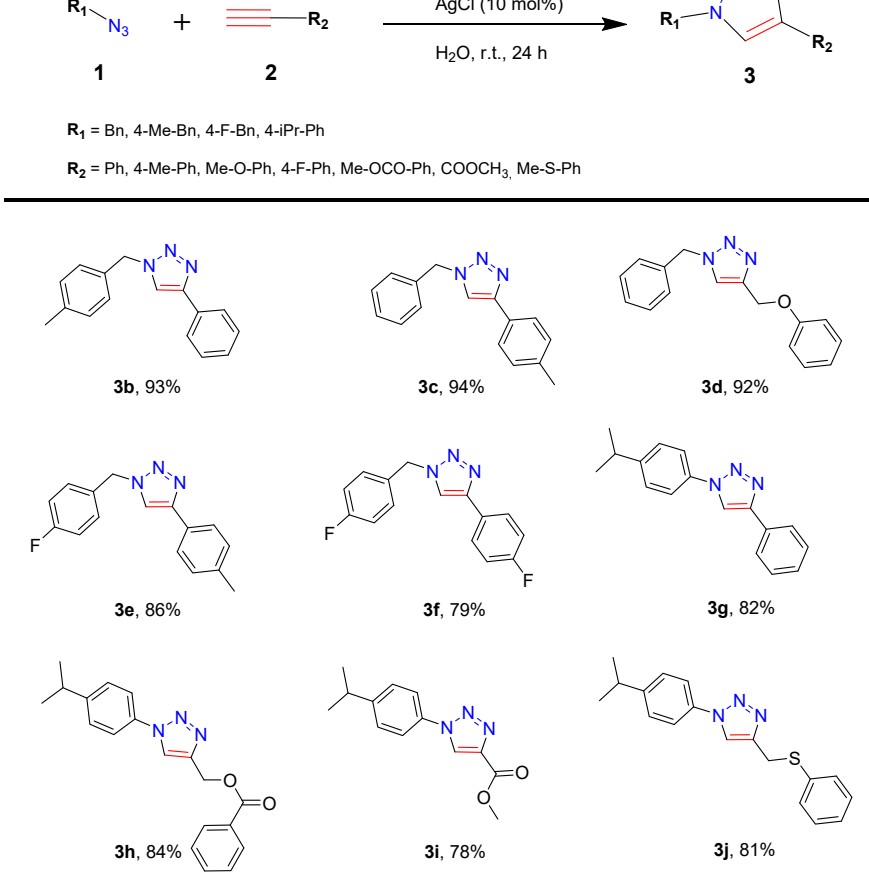

**Scheme 1.** Ag(I)-catalyzed cycloaddition reaction between benzyl azide (**1a**) and phenylacetylene (**2a**).

In order to enlarge the scope of this catalytic 32CA reaction with AgCl as catalyst, several 1,4-disubstituted-1,2,3-triazoles were prepared in good to high yields, including examples bearing electron-releasing or electron-withdrawing groups. In all cases, the corresponding triazole derivatives were easily separated from the reaction mixture by the process of extraction without need for purification through column chromatography (Scheme 2). The products were systematically characterized by their melting points determination, $^1$H and $^{13}$C NMR spectroscopy and mass spectrometry (see Supplementary Materials for details).

**Scheme 2.** 1,2,3-triazole derivatives prepared under AgAAC reaction conditions.

In order to prove the utility and potential of our synthetic method using AgCl as catalyst in AAC reactions, a comparison with other AgAAC processes was done (see Table 1). It is clear that the presence of a ligand and the performance of the 32CA reaction at high temperatures are prerequisites for some catalytic systems to be operative. In contrast, the use of aqueous suspensions of AgCl ($K_{ps} = 1.77 \times 10^{-10}$) in the dark at room temperature, followed by a simple separation of the resulting 1,4-disubstituted-1,2,3-triazoles proved to be an operative and strictly clickable approach.

**Table 1.** Comparative study of AgAAC reactions of phenylacetylene (**1a**) and benzyl azide (**2a**) under different reaction conditions.

| Entry | Ag-Cat. | Conditions | Yield (%) | Ref. |
|:-----:|:-------:|:----------:|:---------:|:----:|
| 1 | AgCl (10 mol%) | $H_2O$/r.t./24 h | 92 | This work |
| 2 | AgCl (5 mol%) | $H_2O$/acetone/r.t./24 h | 64 | [25] |
| 3 | AgCl (20 mol%) | THF/60 °C/TEA/4 h | 87 | [35] |
| 4 | $AgN(CN)_2$ (10 mol%) | $H_2O$/EG/DIPEA/2 h | 98 | [27] |
| 5 | AgCl (0.1 mol%)/**L1** | THF/r.t./15 h | 77 | [25] |
| 6 | $Ag(OOCCH_3)$ (20 mol%)/**L2** | Toluene/r.t./Caprylic acid/48 h | 98 | [23] |
| 7 | $Ag(OOCCH_3)$ (2 mol%)/**L3** | Toluene/90 °C/Caprylic acid/24 h | 98 | [24] |
| 8 | $Ag_2CO_3$ (10 mol%) | $H_2O$/CPyCl/r.t./2 h | 98 | [36] |

**L1** = 1,3-bis-(2,6-diisopropylphenyl)-2,4-diphenylimidazolium chloride. **L2** = *N,N*-diisopropyl-(2-diphenylphosphanyl)benzamide. **L3** = *N,N*-Diisopropyl-2-(di-*tert*-butylphosphanyl)benzamide. CPyCl = Cetylpyridinium chloride.

## 2.2. MEDT Study

Cu(I)-acetylide has been suggested as the beginning species of the catalytic cycle in CuAAC reactions where the alkyne substrate first binds to copper(I) in a π-coordination mode. In fact, the acidity of the terminal alkyne proton is increased as a consequence of the formation of stable µ-acetylide copper(I) intermediates [37]. The subsequent formation of a van der Waals complex between the metal-acetylide species and azide derivative, which has already been addressed in several Ru-, Cu-, and Zn-mediated cycloaddition reactions, was found to depend on the type of metal, and ligands as well as on reaction conditions [38–41]. Moreover, the long-term debate on the number of copper centers participating in the CuAAC reaction after theoretical and experimental studies pointed to the fact that the second participating copper(I) ion facilitates the generation of the cupracycle in the rate-determining step (RDS) and stabilizes the metallacycle intermediate itself [40,42,43]. Cu(I)-acetylides can also be formed via a transmetallation process with Ag(I)-acetylide precursors and subsequently reacted with organic azides to produce 1,4-disubstitued- 1,2,3-triazoles, whereas in absence of copper(I), the product could not be obtained by using only Ag(I)-acetylides [44]. The fact that the reactions proceed in water can be explained by the great affinity of silver(I) ions and alkynes resulting in the generation of the Ag(I)-acetylide precursor.

Based on this background, we considered that the formation of Ag(I)-acetylide structures with a water molecule as ligand is the starting point of the AgAAC reaction path that affords 1,4-disubtituted-1,2,3-triazoles. A different number of silver(I) centers, namely mono- and dinuclear Ag(I)-acetylides, were also modelled based on recent theoretical studies by analogy with the reactivity of Cu(I)-acetylides (Figure 2) [32,42].

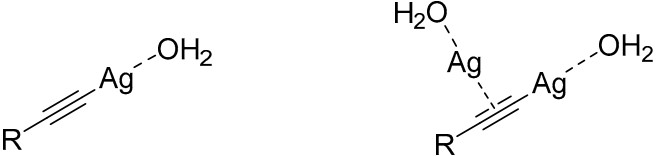

Mononuclear Ag(I)-Acetylide **(Ac1)**     Binuclear Ag(I)-Acetylide **(Ac2)**

**Figure 2.** Formation of Ag(I)-acetylide species in water.

In the proposed mechanism of AgAAC reactions shown in Figure 1, the first step proceeds via the formation of metallated terminal alkyne that involves one Ag(I) atom, by an easy deprotonation of the alkyne precursor, yielding the Ag(I)-acetylide species **Ac1** (see Figure 2). The next step consists in the coordination of the azide precursor to the silver(I) of the species **Ac1**, forming the cycloaddition

adduct, which subsequently yields the corresponding 1,2,3-triazole derivatives. Herein, the mono- and dinuclear reaction paths associated to the AgAAC reactions of propyne (**4**) with methyl azide (**5**) affording 1,4-dimethyl-1,2,3-triazole (**6**) are investigated through MEDT, using DFT methods at the B3LYP/6-31g(d,p) (LANL2DZ for Ag) computational level (see Figure 3). Two structures have been considered for the dinuclear Ag(I)-acetylide: each one of the two Ag(I) cations in **Ac2** has a coordinated water molecule, while in **Ac3**, one of the two silver(I) cations has a chloride ligand. Thus, while **Ac1** and **Ac3** are neutral species, **Ac2** is a cationic one (see Figure 4).

**Figure 3.** AgAAC reaction between **4** and **5**.

**Figure 4.** Possible Ag(I)-acetylide intermediates.

### 2.2.1. Analysis of the Global and Local CDFT Reactivity Indices

Azides are linear three-atom-components (L-TACs) participating in *zw*-type 32CA reactions with high activation energies [45], which can be more favored via a polar process by the increase of the electrophilic and nucleophilic character of the reagents. Accordingly, the AgAAC reactions under study were analyzed using the global reactivity indices defined within conceptual DFT (CDFT) [46,47]. The global indices, namely, the electronic chemical potential ($\mu$) chemical hardness ($\eta$) electrophilicity ($\omega$), and nucleophilicity (*N*) of the reagents are given in Table 2.

**Table 2.** Global reactivity indices, namely, $\mu$, $\eta$, $\omega$ and *N* (in eV) of the reagents involved in AgAAC reactions.

| Species | $\mu$ | $\eta$ | $\omega$ | *N* |
|---|---|---|---|---|
| Propyne (**4**) | −2.69 | 8.72 | 0.41 | 2.06 |
| Methyl azide (**5**) | −3.85 | 6.19 | 1.20 | 2.17 |
| Mononuclear Ag(I)-acetylide (**Ac1**) | −3.49 | 4.04 | 1.51 | 3.61 |
| Dinuclear Ag(I)-acetylide (**Ac2**) | −7.70 | 5.17 | 5.73 | −1.16 |
| Dinuclear Ag(I)-AgCl-acetylide (**Ac3**) | −4.09 | 3.86 | 2.16 | 3.10 |

In a polar AAC reaction, a global electron density transfer (GEDT) takes place from the nucleophilic fragment toward the electrophilic one, whose direction can be established according to the equalization principle of the electronic chemical potential [48]. The electronic chemical potential of **4** ($\mu$ = −2.69 eV) is greater than that of **5** ($\mu$ = −3.85 eV) pointing to the fact that GEDT should take place from **4** to **5** along a polar 32CA reaction, thus acting as nucleophile and electrophile, respectively. Furthermore, the electronic chemical potentials of the neutral acetylides **Ac1** and **Ac3** (−3.49 and −4.09 eV), respectively) are close to that of **5**, indicating that the AgAAC reactions of **Ac1** and **Ac3** with **5** should present a low polar character. Conversely, the electronic chemical potential of **Ac2** ($\mu$ = −7.62 eV) is very low because of the cationic nature of this species. Interestingly, the electronic chemical potentials

of **Ac2** and **Ac3** are lower than that of **5**, which indicates that **Ac2** and **Ac3** are more likely to act as the electrophiles instead of **Ac1** in the corresponding AgAAC reactions.

    **4** has $\omega$ and $N$ indices of 0.41 and 2.06 eV respectively, being classified as a marginal electrophile and a marginal nucleophile within the electrophilicity [49] and nucleophilicity scales [50]. These very low values explain the non-participation of **4** in polar cycloaddition reactions. On the other hand, **5** is a strong electrophile ($\omega$ = 1.20 eV) and a moderate nucleophile ($N$ = 2.17 eV).

    The coordination of silver(I) to the terminal C5 carbon of **4** increases the values of the $\omega$ index of the corresponding mono- and dinuclear Ag(I)-acetylides to 1.51 (**Ac1**), 5.73 (**Ac2**), and 2.16 eV (**Ac3**), being classified as strong electrophiles. Nevertheless, their $N$ index increases to 3.61 (**Ac1**) and 3.16 (**Ac3**) at the neutral acetylides, then being classified as strong nucleophiles, but it strongly decreases to −1.16 eV for **Ac2** because of the cationic character of this species.

    Therefore, unlike the 32CA reaction of **5** with **4**, which will have a non-polar character due to the poor nucleophilic activation of **4**, the AgAAC reactions of **5** with **Ac1**, **Ac2**, and **Ac3** will exhibit some polar character due to the strong electrophilic character of **Ac2** and **Ac3** and the strong nucleophilic character of **Ac1.**

    The most favorable regioisomeric reaction path in polar 32CA reactions is that involving the two-center interaction between the strongest electrophilic and the strongest nucleophilic centers of the reagents. Several studies of 32CA reactions, especially the reactions with a strong polar character, have shown that the experimentally observed regioselectivity can be explained by the analysis of the electrophilic and nucleophilic Parr functions [48]. Thus, the nucleophilic $P_k^-$ and electrophilic $P_k^+$ Parr functions at **4**, **5** and **Ac1**, **Ac2**, and **Ac3** were analyzed (see Figure 5).

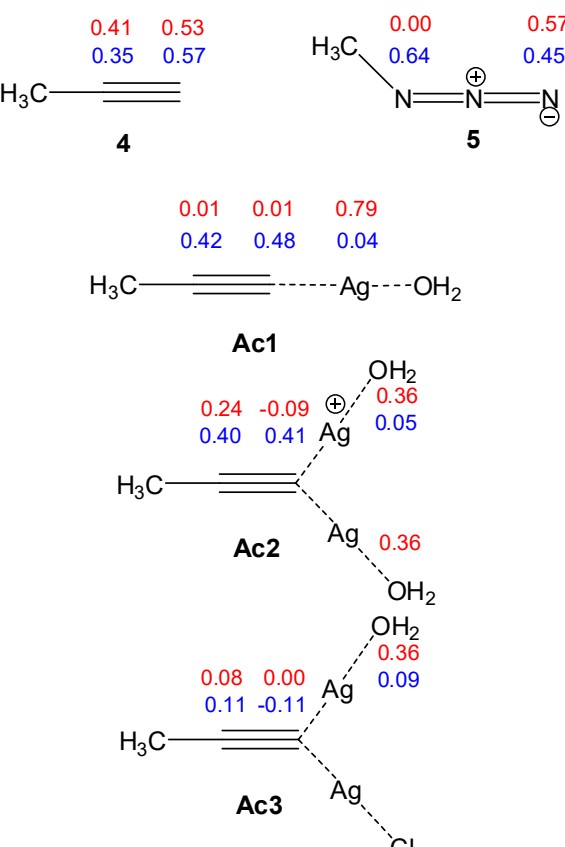

**Figure 5.** Electrophilic $P_k^+$ Parr (red) and nucleophilic $P_k^-$ Parr (blue) functions at **4**, **5** and **Ac1**, **Ac2**, and **Ac3**.

    The analysis of the local reactivity indexes using Parr functions indicates that the terminal C5 carbon at **4** is the most nucleophilic center of this molecule ($P_k^-$ = 0.57), while the C4 carbon is also nucleophilically activated ($P_k^-$ = 0.35). On the other hand, the terminal N1 nitrogen is the most

electrophilic center of this TAC ($P_k^+ = 0.57$), which suggests that the formation of 1,5-regioisomer is more favorable than 1,4-regioisomer along a polar 32CA reaction. The very low polar character of the chosen uncatalyzed AAC reaction accounts for its low regioselectivity.

Figure 5 illustrates local reactivity indexes, the electrophilic $P_k^+$ and nucleophilic $P_k^-$ Parr functions, of **Ac1**, **Ac2**, and **Ac3**. According to the direction of the GEDT suggested by the electronic chemical potentials, a similar local reactivity to that of **4** is predicted for **Ac1**. Otherwise, the electrophilic Parr functions at **Ac2** and **Ac3** characterize the silver atom interacting with a water molecule as the most electrophilic center ($P_k^+ = 0.36$ ), whereas the substituted N3 nitrogen is the most nucleophilic center ($P_k^- = 0.64$) as found by the analysis of the nucleophilic Parr functions at **5**. Therefore, unlike the low-polar reactions involving **4** and **Ac1**, a high N3-C5 regioselectivity favoring the formation of 1,4-disubstituted-1,2,3-triazoles will be expected in the polar AgAAC reactions of **Ac2** and **Ac3**. It is worth mentioning that Parr functions allow the characterization of the most favorable regioisomeric reaction path in polar reactions under kinetic control; however, these AgAAC reactions start with the formation of a reactive complex, which determines the regioselectivity of these AgAAC reactions (*vide-infra*).

### 2.2.2. Mononuclear Mechanism

The mononuclear AgAAC mechanism starting from **Ac1**, in which one water molecule participates as a ligand was first studied (see Figure 6). The coordination of the substituted N3 nitrogen of **5** to the Ag(I) cation of **Ac1** occurs at the initial step. In one elementary step, the coupling of **5** with **Ac1** forms the metallated Ag(I)-triazolide **AT1** via **TS1** (see Figure 6).

**Figure 6.** AgAAC reaction of **Ac1** with **5**.

The geometries of the stationary points involved in the two regioisomeric reaction paths associated with the mononuclear AgAAC reaction of **Ac1** with **5** are shown in Figure 7. The distance between the C5 carbon and the Ag(I) cation in **Ac1** is 2.043 Å, a value which is slightly larger than that found at the Ag(I)-acetylide using PH$_3$ as ligand (2.03 Å) [32]. In the case of **RC1**, the distances between the silver(I) cation and the N3 nitrogen and the C5 carbon atoms are 2.919 and 2.054 Å, respectively. Concerning **TS1**, the N1-C4 distance is 1.804 Å, while those between the silver(I) cation and the N3 nitrogen and the C5 carbon atoms are 2.385 and 2.056 Å, respectively. At this TS, the distance between the N3 nitrogen and the C5 carbon atoms is 2.978 Å. This value together with that of the N1-C4 distance show the high asynchronous character of **TS1**, which is associated with a *two-stage one-step* mechanism (note the great similitude of **TS1** with the TSs of the first step at the mechanisms of the reactions of dinuclear Ag(I)-acetilydes in Figure 7). Finally, the distance between the C5 carbon atom and the silver(I) cation at **AT1** is 2.089 Å. The analysis of the geometrical parameters of the species involved in the regioisomeric reaction path shows a great similitude; only in **RC1r**, the N1-Ag(I) distance (3.611 Å) is longer than that in **RC1**, as a consequence of a weaker azide-Ag(I) interaction.

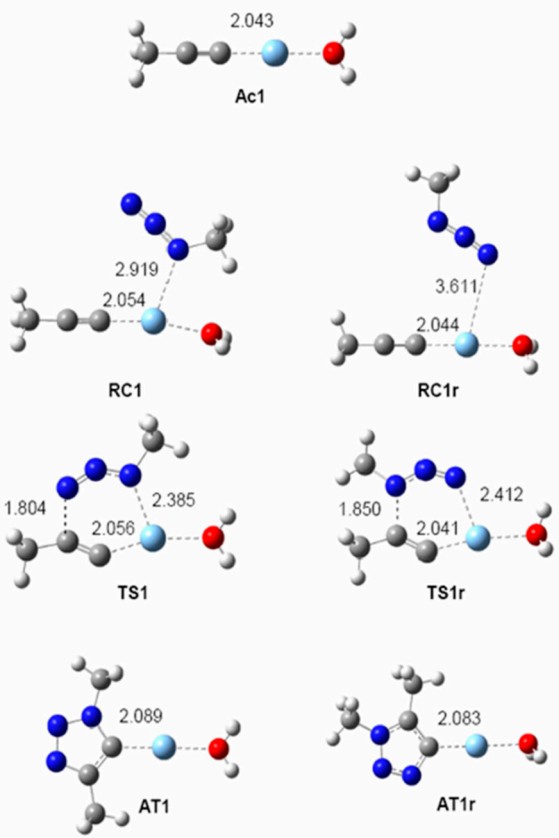

**Figure 7.** Optimized geometries (in water) of the stationary points associated with the mononuclear AgAAC reaction of **Ac1** with **5**. Distances are given in Å.

The reactive complex **RC1** is found 1.6 kcal/mol above the non-interacting reagents, **Ac1** and **5** (Figure 8). From the separated reagents, the activation energy associated with the formation of the metallated Ag(I)-triazolide **AT1** via **TS1** is 21.8 kcal/mol as shown in Figure 8. This activation energy of 1.5 kcal/mol was found to be above the one associated with the non-catalyzed process (see Supplementary Materials for details), led us to formulate the hypothesis of the dinuclear mechanism pathway. The formation of **AT1** is strongly exothermic by 60.9 kcal/mol as shown in Figure 8. At the end of the process, the Ag(I) catalyst can be detached by protonation.

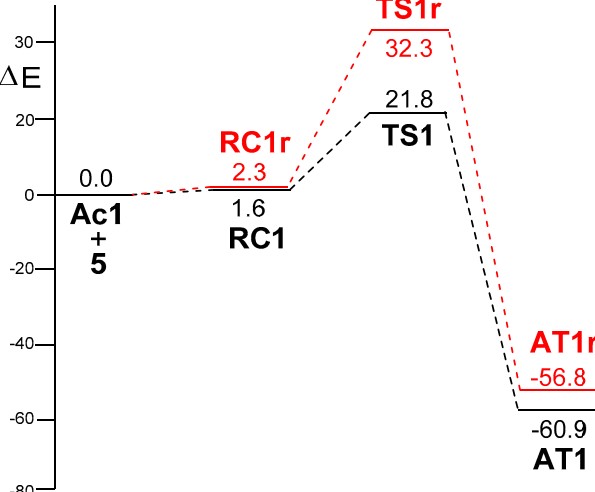

**Figure 8.** Energy profiles (in kcal/mol) of the two regioisomeric reaction paths associated with the mononuclear AgAAC reaction of **Ac1** with **5**.

Interestingly, while the 32CA reaction of azides are poorly regioselective, the AgAAC ones are completely regioselective yielding only 1,4-triazoles. With a view to explain the origin of the regioselectivity, the reaction path allowing the formation of 1,5-triazole **AT1r** was also studied as illustrated in Figure 8. Such a formation of **RC1r** where the terminal N1 nitrogen of azide is coordinated to the Ag(I) species is only 0.7 kcal/mol above that of **RC1**. From the non-interacting reagents, the activation energy associated to the formation of **AT1r** through **TS1r** is 32.3 kcal/mol. The formation of **AT1r** is 4.1 kcal/mol less exothermic than that of **AT1**. Consequently, the pathway to **AT1** via **TS1** is 10.5 kcal/mol kinetically more favorable than that of **AT1r** via **TS1r**, in full agreement with the total regioselectivity found in the AgAAC reactions.

The polar nature of the mononuclear AgAAC reaction was analyzed evaluating the GEDT at **RC1** and **TS1**. The values of the GEDT are −0.02 and 0.28 e at **RC1** and **TS1**, respectively. There is an unappreciable GEDT taking place from the azide to the acetylene at the reactive **RC1** complex, but when the 32CA reaction begins, the electron density moves from the acetylide framework to the azide one, in good agreement with the CDFT reactivity indices. The GEDT found at **TS1** is indicative of the polar character of this AgAAC reaction. Note that the GEDT at the TS associated with the non-catalyzed 32CA reaction is negligible (ca. 0.04 e), indicating the non-polar nature of the reaction. At **TS1r**, the GEDT is 0.14 e; this smaller value when compared with that of **TS1** accounts for the lower polar character of the former and the higher activation energy.

### 2.2.3. Dinuclear Mechanism

Analysis of the stationary points involved in the AgAAC reactions of the dinuclear **Ac2** or **Ac3** species with **5** indicates that these reactions take place through stepwise mechanisms (see Figure 9). The geometries of the **TSs** and the intermediate involved in the dinuclear AgAAC reactions of **Ac2** and **Ac3** with **5** are given in Figure 10, while the energy profiles of the reaction pathways associated with the AgAAC reactions of **Ac2** and **Ac3** with **5** are represented in Figure 11. As one can figure out, the species involved in the two reaction paths have similar relative energies, indicating that the substitution of the coordinated water molecule at the second silver(I) cation by a chloride anion does not produce any remarkable change.

**Figure 9.** AgAAC reactions of the dinuclear **Ac2** or **Ac3** species with **5**.

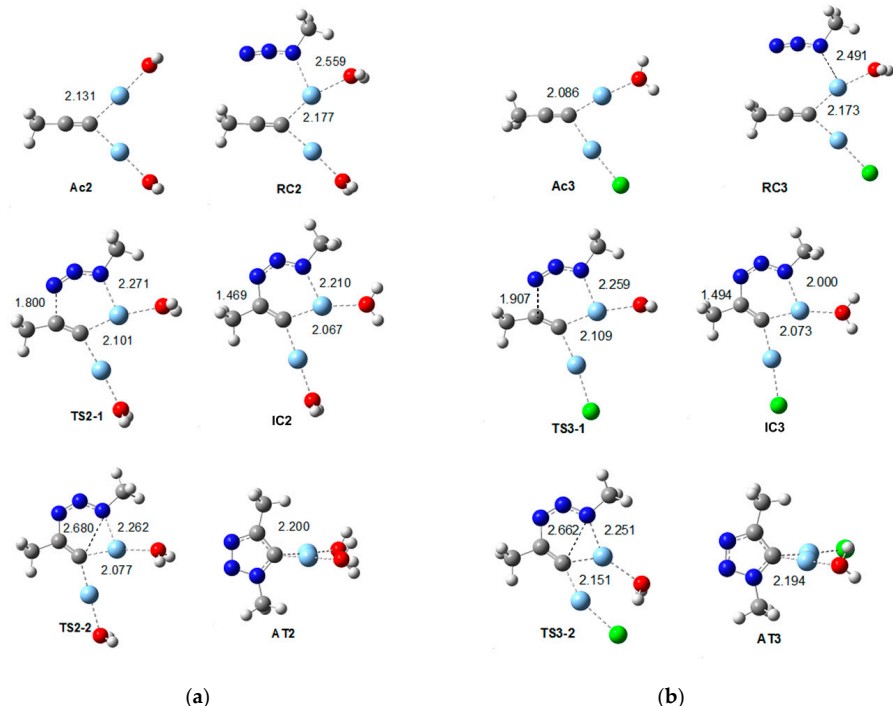

**Figure 10.** Optimized geometries (in water) of the stationary points involved in the dinuclear AgAAC reaction of (**a**) **Ac2** and (**b**) **Ac3** with **5**. Distances are given in Å.

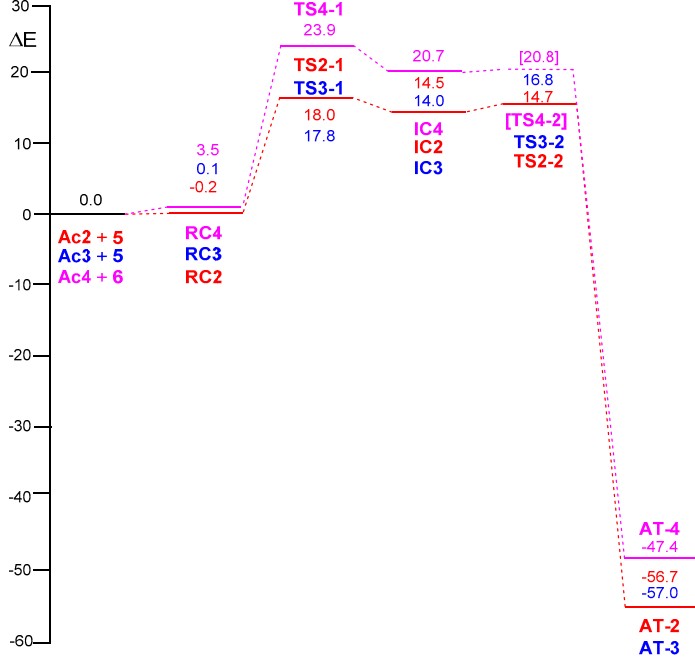

**Figure 11.** Energy profiles (in kcal/mol) of the most favorable regioisomeric reaction paths associated with the AgAAC reactions of **Ac2** (in red) and **Ac3** (in blue) with **5**, and that of **Ac4** (in pink) with **6**.

A comparison of the geometric parameters of the species involved in the two reaction paths indicates that the substitution of a water molecule by the chloride anion does not produce any remarkable change, in a clear agreement with the similar relative energies (Figure 10). At the dinuclear Ag(I)-acetylides, the C5-Ag distances are 2.138 (**Ac2**) and 2.086 Å (**Ac3**) in agreement with previous theoretical studies using $PH_3$ as a ligand (2.13 Å) [32]. At the reactive complexes associated with the nucleophilic attack of the azide N3 nitrogen atom to the dinuclear Ag(I)-acetylides, the distances

between N3 and Ag are 2.559 (**RC2**) and 2.491 Å (**RC3**). At the first **TS** of these stepwise mechanisms, the lengths of the N1-C4 forming bond are 1.800 (**TS2-1**) and 1.907 Å (**TS3-1**), while the distances between the N3 nitrogen atom and Ag(I) are 2.271 (**TS2-1**) and 2.259 Å (**TS3-1**). At the intermediate complexes, the lengths of the N1-C4 bond are 1.469 (**IC2**) and 1.494 Å (**IC3**), while the N3-Ag distances are 2.210 (**IC2**) and 2.000 Å (**IC3**). Finally, at the second **TSs** associated with the ring-closure process, the lengths of the N3-C5 forming bonds are 2.680 (**TS2-2**) and 2.662 Å (**TS3-1**).

The first step of the dinuclear AgAAC consists in the formation of mononuclear Ag(I)-acetylide intermediate **Ac1** via the coordination of silver(I) cation and **4** as a $\pi$-type ligand. The subsequent coordination of the C5 carbon atom of **Ac1** to a second silver(I) center yields two feasible **Ac2** or **Ac3** dinuclear species depending on the ligand attached to the second Ag(I) cation, i.e., a water molecule or a chloride anion. Thus, the coordination of **5** to a silver(I) cation in **Ac2** or **Ac3** via its azide N1 nitrogen atom permits the formation of the reactive **RC2** and **RC3** complexes, respectively. These species represented in Figure 10 are viewed as starting points for the stepwise sequences. In water, these steps are practically isothermal; −0.2 (**RC2**) and 0.1 kcal/mol (**RC3**). The computed barriers for the formation of the first N1-C4 single bond via **TS2-1** or **TS3-1** are 18.0 and 17.8 kcal/mol, respectively, values which are very low compared to the value found in the non-catalyzed type-reaction (20.3 kcal/mol), and ca. 2 kcal/mol below that found for the AgAAC reaction involving the mononuclear Ag(I)-acetylide **Ac1**. The formation of the intermediate complexes **IC2** and **IC3** is endothermic by ca. 14 kcal/mol. From these intermediates, the values of the energy barrier for the ring contraction leading to the Ag(I)-triazolyl derivatives are 0.2 (**TS2-2**) and 2.8 kcal/mol (**TS3-2**). Finally, the formation of the **AT2** and **AT3** derivatives is exothermic by 56.7 and 57.0 kcal/mol, respectively.

The polar nature of the dinuclear AgAAC reaction was analyzed evaluating the GEDT at the reactive complexes, intermediates, and **TSs**. Along the two reaction paths, the values of the GEDT are −0.02, −0.05, 0.16, 0.18, 0.36, and 0.38 e at **RC2**, **RC3**, **TS2-1**, **TS3-1**, **IC2**, and **IC3**, respectively. At the reactive complexes, the GEDT from the azide to the acetylene frameworks is very low, but subsequently, the electron density fluxes from the acetylide framework to the methyl azide one. Nevertheless, the calculated GEDT values indicate that these AgAAC reactions have a polar character.

In order to validate the B3LYP functional, the AgAAC reaction of the dinuclear **Ac2** with **5** was studied by using the ωB97XD functional together with the 6-311G(d,p) basis set for C, N, O, and H nuclei and the LANL2DZ type basis set for Ag. The energy results and the TSs geometries are given in the Supporting Material. As show Figure S1, ωB97XD functional gives a similar stepwise mechanism to that obtained by using the B3LYP one. The more remarkable difference is a greater stabilization of **RC2**, −7.5 kcal/mol, and **AT3**, −70.5 kcal/mol with the ωB97XD functional. **TS1-2**, **IC-2,** and **TS2-2** are found only between 2 and 3 kcal/mol below the B3LYP ones. The ωB97XD geometries of the TSs given in Figure S3 show that there is a great similitude between the B3LYP and ωB97XD geometries of **TS1-2** and **TS2-2**, the ωB97XD TSs being slightly more advanced. This comparative analysis validates the use of B3LYP functional for the study of these AgAAC reactions.

Finally, in order to validate the reduced model used in this MEDT study, the AgAAC reaction of the dinuclear **Ac4** with phenyl azide **6** was also studied (see Figure 12). Because of the very flat character of the potential energy surface around **IC4** and **TS4-2**, it was not possible to find **TS4-2** as a stationary point. In any case, both AgAAC reactions take place through a stepwise mechanism. The energy results are given in Figure 11. As one can see therein, all species involved in the AgAAC reaction of the dinuclear **Ac4** with **6** are between 4 and 9 kcal/mol higher in energy that those involving dinuclear **Ac2** and **5**. Thus, the activation energy associated to the reaction of **Ac4** with **6**, via **TS4-1**, is found 5.7 kcal/mol higher in energy than that associated to the reaction of **Ac2** with **5**.

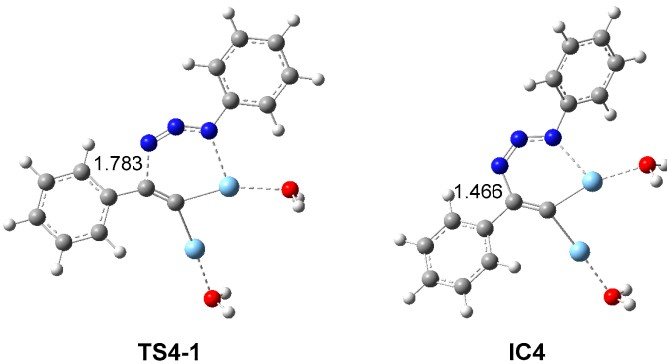

**Figure 12.** AgAAC reactions of the dinuclear **Ac4** species with **6**.

The geometries of **TS4-1** and **IC4** associated to the dinuclear AgAAC reaction of **Ac2** and **Ac4** with **6** are given in Figure 13. At **TS4-1**, the distance between two C–N interacting centers (1.783 Å) is slightly shorter than that at **TS2-1**, in agreement with the higher energy of the former entity. The GEDT at **TS4-1** and **IC4** are 0.18 and 0.50 e respectively, indicating that this AgAAC reaction has a polar character.

TS4-1             IC4

**Figure 13.** Optimized geometries (in water) of **TS4-1** and **IC4** associated to the dinuclear AgAAC reaction of **Ac2** and **Ac4** with **6**. Distances are given in Å.

## 2.3. ELF Topological Analysis of the Stationary Points Involved in the AgAAC Dinuclear Reaction of Ac3

In order to shed light onto the molecular mechanism of the AgAAC dinuclear reaction of **Ac3**, the electronic structure of the corresponding stationary points was characterized by a topological analysis of the ELF. Within the context of ELF, monosynaptic valence basins [labelled V(A)] are associated with non-bonding regions, i.e., lone pairs or *pseudoradical* centers, while disynaptic valence basins [noted V(A,B)] connect the core of two nuclei A and B and thus, correspond to a bonding region between A and B [51]. This description, together with the ELF valence basin populations, provide a straightforward connection between the electron density distribution and the chemical structure as represented by the Lewis's bonding model. The populations of the most relevant ELF valence basins associated to the reagents and stationary points are gathered in Table 3 and Figure 14, together with ELF-based Lewis-like representations of their electronic structure, while the ELF basin attractor positions are shown in Figure 15.

**Table 3.** ELF valence basin populations, values of the distances of the forming bonds and global electron density transfer (GEDT) of the stationary points involved in the dinuclear *zw*-type AgAAC reaction of **Ac3** with **5** [the values of the distances are given in Å and those of the GEDT and electron populations in average number of electrons (e)].

| Bond Length | 5 | Ac3 | RC3 | TS3-1 | IC3 | TS3-2 | AT3 |
|---|---|---|---|---|---|---|---|
| d(N1-C4) | - | - | 4.246 | 1.808 | 1.467 | 1.445 | 1.362 |
| d(N3-Ag) | - | - | 2.555 | 2.277 | 2.219 | 2.251 | 3.108 |
| d(Ag-C5) | - | 2.134 | 2.178 | 2.101 | 2.074 | 2.157 | 2.194 |
| d(N3-C5) | - | - | 4.041 | 3.053 | 2.793 | 2.662 | 1.381 |
| GEDT | - | - | −0.05 | 0.17 | 0.38 | 0.42 | 0.31 |
| V(N1,N2) | 1.74 | - | 1.79 | 2.49 | 2.17 | 2.12 | 1.87 |
| V'(N1,N2) | 2.26 | - | 1.79 | - | - | - | - |
| V(N2) | - | - | - | 2.29 | 2.65 | 2.70 | 3.24 |
| V(N2,N3) | 2.50 | - | 2.43 | 1.76 | 1.75 | 1.80 | 1.66 |
| V(C4,C5) | - | 2.48 | 2.41 | 2.10 | 1.75 | 1.74 | 2.93 |
| V'(C4,C5) | - | 2.32 | 2.41 | 2.03 | 1.83 | 1.71 | - |
| V(N1) | 3.94 | - | 3.89 | 4.05 | 3.36 | 3.35 | 3.07 |
| V(N3) | 3.53 | - | 3.59 | 1.81 | 1.70 | 3.50 | 0.77 |
| V'(N3) | - | - | - | 1.82 | 1.85 | - | 0.75 |
| V(C4) | - | - | - | 0.26 | - | - | - |
| V(C5) | - | 2.91 | 2.91 | 3.16 | 3.24 | 3.19 | 2.68 |
| V(N1,C4) | - | - | - | - | 1.64 | 1.75 | 2.29 |
| V(N3,C5) | - | - | - | - | - | - | 2.08 |

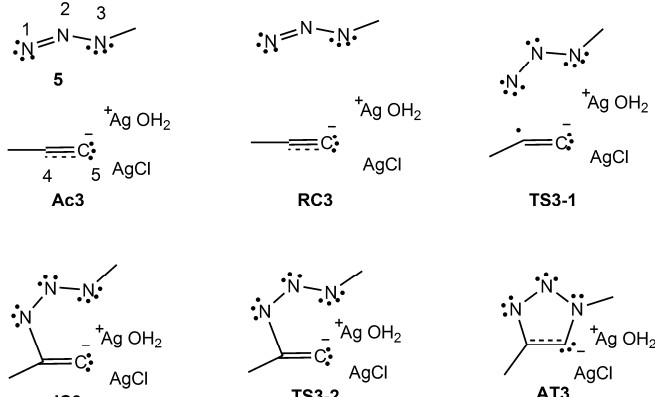

**Figure 14.** ELF-based Lewis-like structures of the stationary points involved in the dinuclear *zw*-type AgAAC reaction of **Ac3** with **5**.

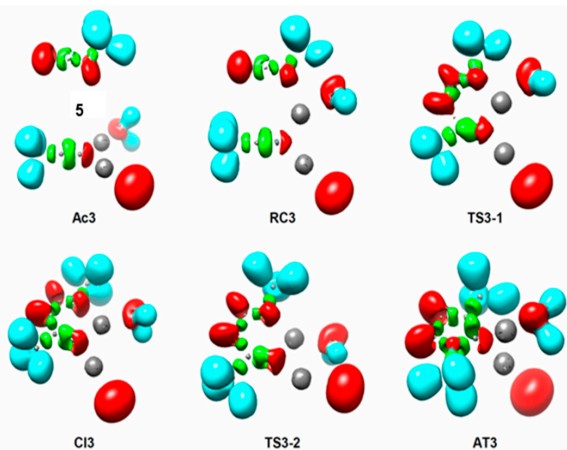

**Figure 15.** ELF localization domains (represented at an ELF value of 0.75) of the stationary points involved in the binuclear *zw*-type AgAAC reaction of **Ac3** with **5**. ELF localization domains of **TS3-1** were represented at ELF = 0.68 to show the V(C4) monosynaptic basin.

The topological analysis of the ELF of **5** shows the presence of two V(N1) and V(N3) monosynaptic basins integrating 3.94 and 3.53 e, two V(N1,N2) and V'(N1,N2) disynaptic basins integrating a total population of 4.00 e, and one V(N2,N3) disynaptic basin integrating 2.50 e. According to the Lewis' bonding model, these valence basins can be related to two lone pairs at both terminal N1 and N3 azide nitrogen nuclei, one N1-N2 double, and a N2-N3 single bond (see the ELF-based Lewis-like structures in Figure 14). Consequently, **5** can be characterized as a zwittterionic TAC participating in *zw*-type 32CA reactions [52]. On the other hand, the electronic structure of **Ac3** is characterized by the presence of two V(C4,C5) and V'(C4,C5) disynaptic basins, integrating a total population of 4.80 e, and one V(C5) monosynaptic basin, integrating a high electron population of 2.91 e. These valence basins, which can be associated with an overpopulated C4-C5 double bond and an overpopulated C5 carbanionic center, reveal the ionic nature of **Ac3** (see Table 3).

No appreciable topological changes with respect to the ELF topological characterization of the separated reagents are observed at **RC3** where d(N1-C4) = 4.246 Å and d(N3-C5) = 4.041 Å. Only slight electron density rearrangements are observed. For example, in the azide framework, while the V(N2,N3) disynaptic basin is slightly depopulated by 0.07 e, the total population of the V(N1,N2) and V'(N1,N2) disynaptic basins increases by an equivalent amount. A similar behavior is found for the two V(N1) and V(N3) monosynaptic basins. The GEDT fluxing from the azide to the acetylide frameworks at **RC3** is very low, ca. 0.05 e.

Several relevant topological changes occurred at **TS3-1** where d(N1-C4) = 1.808 Å and d(N3-C5) = 3.053 Å. On the one hand, in the azide framework, while the total population of the two V(N1,N2) and V'(N1,N2) disynaptic basins, as well as that of the V(N2,N3) disynaptic one, have strongly decreased by 1.58 and 0.67 e respectively, a new V(N2) monosynaptic basin, associated with the N2 nitrogen non-bonding electron density, has been created integrating 2.29 e at **TS3-1**. In addition, the V(N3) monosynaptic basin present in **RC3** has split into two V(N3) and V'(N3) monosynaptic basins integrating 1.81 and 1.82 e, although their total population has only increased by 0.04 e. Otherwise, the population of the V(N1) monosynaptic basin has more markedly increased to 4.05 e, but the N1 nitrogen non-bonding electron density remains characterized by one single V(N1) monosynaptic basin. At the other hand, in the acetylide framework, together with the strong depopulation of the two V(C4,C5) and V'(C4,C5) disynaptic basins by a total of 0.69 e, the V(C5) monosynaptic basin has reached 3.16 e and a new V(C4) monosynaptic basin has been created integrating 0.26 e at **TS3-1**, which can be related to a C4 *pseudoradical* center. The GEDT taking place from the acetylide frameworks toward the azide one has significantly increased to 0.17 e at **TS3-1**.

At **IC3** where d(N1-C4) = 1.467 Å and d(N3-C5) = 2.793 Å, a new V(N1,C4) disynaptic basin, integrating 1.64 e is observed together with the disappearance of the V(C4) monosynaptic basin present at **TS3-1** and the depopulation of the V(N1) monosynaptic basin by 0.69 e. These topological changes suggest that the V(N1,C4) disynaptic basin, which is associated to the new N1-C4 single bond, has been formed by sharing part of the electron population of the V(N1) monosynaptic basin and that of the V(C4) one (see Figure 14). In addition, the two V(C4,C5) and V'(C4,C5) disynaptic basins have been considerably depopulated by a total of 0.63 to 3.50 e, which probably will have contributed to the further population of the V(N1,C4) disynaptic basin after its creation, as well as to the slight increase of the population of the V(C5) monosynaptic basin to 3.24 e. In the azide framework, the population of the V(N2) monosynaptic basin has continuously increased to 2.65 e as that of the V(N1,N2) disynaptic basin has decreased to 2.17 e. The population of the V(N2,N3) disynaptic basin remains, however, unchanged. At **IC3**, the GEDT has strongly increased to 0.38 e.

d(N1-C4) = 1.445 Å and d(N3-C5) = 2.662 Å are found at **TS3-2** and the most relevant topological change with respect to the ELF topological characterization of **IC3** is the merger of the two V(N3) and V'(N3) monosynaptic basins into one single V(N3) monosynaptic basin, integrating 3.50 e, as a consequence of the slight depopulation of 0.05 e toward the V(N2,N3) disynaptic basin, whose population increases to 1.80 e. This change is, however, chemically meaningless. It is also worth

mentioning that the population of the V(C5) monosynaptic basin, which had been increasing during the reaction progress, has decreased to 3.19 e. The GEDT at **TS3-2**, reaches the highest value, 0.42 e.

Finally, at **AT3** where d(N1-C4) = 1.362 Å and d(N3-C5) = 1.381 Å are found, a new V(N3,C5) disynaptic basin integrating 2.08 e is observed. The creation of this disynaptic basin, which is associated to the new N3-C5 single bond, is accompanied by a notable depopulation of the V(N3) and V(C5) monosynaptic basins by 1.98 and 0.51 e. These population changes suggest that the creation of the V(N3,C5) disynaptic basin has taken place by sharing part of the electron populations of both V(N3) and V(C5) monosynaptic basins, the former in a greater extent. Moreover, because of the planar arrangement of the N3 nitrogen environment, the V(N3) monosynaptic basin present in **TS3-2** has split into two V(N3) and V'(N3) monosynaptic basins integrating a total population of 1.52 e. On the other hand, the V(N1,C4) disynaptic basin has reached 2.29 e, as the V(N1) monosynaptic basin and the two V(C4,C5) and V'(C4,C5) disynaptic basins have been depopulated by 0.28 and 0.52 e respectively, in such a manner that the two last ones have merged into one single V(C4,C5) disynaptic basin integrating 3.07 e. In the azide framework, while the populations of the two V(N1,N2) and V(N2,N3) disynaptic basins have decreased to less than 2 e, the V(N2) monosynaptic basins integrate 3.24 e. The GEDT remains very high at **AT3**, 0.31 e.

Some appealing conclusions can be drawn from this ELF topological analysis: (i) The **Ac3** complex exhibits an ionic nature characterized by the presence of a C5 carbanionic center, which is maintained along the whole reaction path; (ii) the activation energy of the dinuclear AgAAC reaction of **Ac3** via **TS3-1** (17.7 kcal/mol; see Figure 6) can mainly be related to the breaking of the N1-N2 azide double bond, leading to the formation of non-bonding electron density at the N2 nitrogen atom. The N2-N3 and C4-C5 bonding regions are also depopulated but to a lesser extent. This pattern is concordant with the *zw*-type mechanism of 32CA reactions; (iii) the azide N1-N2 bonding region at **TS3-1** can already be considered a single bond, and a C4 *pseudoradical* center resulting from the previous depopulation of the overpopulated acetylide C4-C5 double bond is observed, but neither the N1-C4 nor the N3-C5 single bonds have been still formed (see Figures 14 and 15); (iv) while the formation of the first N1-C4 single bond before reaching **IC3** seems to take place by sharing part of the non-bonding electron density of the azide N1 nitrogen atom and that of the C4 *pseudoradical* center, the formation of the second N3-C5 single bond after **TS3-2** rather corresponds to the donation of part of the N3 non-bonding electron density to the C5 carbanionic center; (v) the populations of the C5 carbanionic center and the N3 nitrogen atom are almost constant along the reaction path until **TS3-2**, suggesting that they only participate in the formation of the second N3-C5 single bond at the end of the reaction (see Table 3); (vi) the electronic structure of **TS3-2** is very similar to that of **IC3**, in agreement with the similar geometries (see Figure 10) and the small activation energy associated with the ring closure, 2.8 kcal/mol (see Figure 11); (vii) and finally, the electronic structure of **AT3** which interestingly does not present a N1-N2 double bond but overpopulated N1 and N2 nitrogen centers instead (see Figure 13), slightly differs from the expected one.

## 3. Materials and Methods

### 3.1. Experimental Details

#### 3.1.1. Reagents and Physical Measurements

All the reagents were purchased from Sigma-Aldrich and Aurora Fine Chemicals. Anhydrous $MgSO_4$ was used for drying organic extracts and all volatiles were removed under reduced pressure. [1]H and [13]C NMR spectra were recorded on a BRUKER DRX-300 AVANCE spectrometer (University of Valencia, Valencia, Spain). NMR spectra were done in $CDCl_3$ as solvent and using tetramethylsilane (TMS) as internal standard. All reaction mixtures were monitored by TLC using commercial glass backed thin layer chromatography (TLC) plates (Merck Kieselgel 60 F254). The plates were observed under UV-light at $\lambda$ = 254 nm. An electrothermal 9100 apparatus was employed to determine the melting points of the final organic products.

### 3.1.2. Typical Procedure for the Ag-Catalyzed Cycloaddition of Alkyne with Azide

AgCl (10 mol%), azide (0.6 mmol), terminal alkyne (0.5 mmol) and water (2 mL) were placed in a round bottom flask and the mixture was stirred at room temperature. The reaction progress was monitored by TLC until the starting materials were consumed. Then, the product was extracted with ethyl acetate (3 × 15 mL) and the combined organic fractions were washed with a saturated $NaHCO_3$ solution (3 × 10 mL) and with brine (20 mL). After drying over $MgSO_4$, the ethyl acetate was removed under reduced pressure and the resulting 1,2,3-triazole derivatives were purely obtained without the need for further purification. The structures of the obtained 1,2,3-triazoles were supported by NMR spectroscopy ($^1$H, $^{13}$C NMR) and mass spectrometry (University of Valencia, Valencia, Spain).

### 3.2. Computational Details

All calculations were carried out using the Gaussian 09 program (G09, Gaussian, Inc., Wallingford CT, 2009) [53]. Geometry optimizations were performed with the B3LYP [54,55] and the ωB97XD [56] functional, together with the 6-31G(d,p) and 6-311G(d,p) basis sets for C, N, O, and H nuclei and the LANL2DZ type basis set for Ag [57,58]. Optimizations were and carried out using the Berny analytical gradient optimization method [59]. The stationary points were characterized by frequency computations in order to verify that TSs have one and only one imaginary frequency. Solvent effects of water were taken into account by full optimization of the gas phase structures at the B3LYP/6-31G(d,p) (LANL2DZ for Ag) computational level using the polarizable continuum model (PCM) developed by Tomasi's group [60] in the framework of the self-consistent reaction field (SCRF) [61–63]. Since some involved species in these AgAAC reactions have a cationic nature, the energy results and geometries are discussed in water.

The GEDT [47] of a reaction is computed by the sum of the atomic charges (*q*) of the atoms belonging to each framework (*f*) at the TSs; GEDT = Σ*qf*. The sign indicates the direction of the electron density flux in such a manner that positive values mean a flux from the considered framework to the other one. The atomic charges were obtained by a natural population analysis (NPA) [64,65].

Topological analysis of the electron localization function (ELF) [66] was performed with the TopMod [67] package using the corresponding B3LYP/6-31G(d,p) (LANL2DZ for Ag) mono-determinantal wave functions and considering a cubical grid of step size of 0.1 Bohr. The molecular geometries were visualized using the GaussView program (Version 6.0, Semichem Inc.: Shawnee, MI, USA, 2009) [68], while the representation of the ELF basin isosurfaces was done by using the UCSF Chimera program (Version 1.11.2, UCSF, California, USA, 2004) [69].

Conceptual DFT [47] (CDFT) provides different indices to rationalize and understand the chemical structure and reactivity. The global electrophilicity index $\omega$ [70], is given by the expression $\omega = (\mu^2/2\eta)$ as a function of the electronic chemical potential ($\mu$) and the chemical hardness ($\eta$) [71]. Both quantities may be approached in terms of the one-electron energies of the frontier molecular orbitals HOMO and LUMO, $\varepsilon_H$ and $\varepsilon_L$, as $\mu \approx (\varepsilon_H + \varepsilon_L)/2$ and $\eta \approx (\varepsilon_L - \varepsilon_H)$, respectively. The global nucleophilicity index *N* [72,73], based on the HOMO energies obtained within the Kohn-Sham scheme [74], is defined as $N = E_{HOMO}(Nu)\text{-}E_{HOMO}(TCE)$, where tetracyanoethylene (TCE) is the reference. CDFT indices were computed from the gas phase B3LYP/6-31G(d,p) (LANL2DZ for Ag), $\varepsilon_H$ and $\varepsilon_L$ energies taken as approximations to the experimental ionization potential and electronic affinity, respectively. The electrophilic $P_k^+$ and nucleophilic $P_k^-$ Parr functions were obtained from the analysis of the Mulliken atomic spin densities (ASD) of the corresponding radical anions and cations of the reagents [75].

## 4. Conclusions

In summary, a combined experimental and MEDT study of the AgAAC reaction was systematically addressed. The present AgAAC protocol involves a fast click of 1,4-disubstituted-1,2,3-triazoles by using AgCl in water and working at room temperature. 1,4-disubstituted-1,2,3-triazoles are obtained in good yields and are easily isolated from the reaction mixture without the need for further purification

methods. From a theoretical viewpoint, the 32CA reaction between **5** and **4** was chosen as the uncatalyzed reaction model, and the AgAAC mechanisms have been studied at both B3LYP/6-31G(d,p) (LANL2DZ for Ag) and the $\omega$B97XD/6-311G(d,p) (LANL2DZ for Ag) computational levels, choosing the water molecule and chloride anion as ligands.

The computational results are correctly explained by means of the more favorable interactions taking place along the 1,4-reaction path. At one hand, while the uncatalyzed 32CA reaction between **4** and **5** presents a high activation energy and a poor regioselectivity, yielding 1,5-disubstituted-1,2,3-triazolide, the AgAAC reactions are favored by 2 kcal/mol, yielding 1,4-disubstituted-1,2,3-triazoles in a regioselective manner, as evidenced by the experimental findings. Indeed, DFT calculations account for the complete regioselectivity of these kinetically controlled AgAAC reactions that afford 1,4-disubstituted-1,2,3-triazoles as the unique product. On the other hand, while the AgAAC reaction of **Ac1** occurs in a single elementary step, the reaction of **Ac2** and **Ac3** follows a two-step mechanism that has no experimental impact due to the endothermic nature of the corresponding intermediate complexes **IC2** and **IC3**. In these cases, the RDS of the reaction is the nucleophilic attack of the azide N1 nitrogen atom to the acetylide C4 carbon atom. The use of chloride or water as ligands does not affect the energetics of the dinuclear reaction paths, avoiding any remarkable competition between them. Moreover, the comparison of the mononuclear and dinuclear AgAAC reaction paths shows that the barriers for the dinuclear processes are ca. 4 kcal/mol lower in energy than that of the mononuclear one. The activation energy of the mononuclear path is higher than the uncatalyzed one [33], which proves that the dinuclear path is more favorable than the mononuclear one.

In order to validate the choice of the computational level, the AgAAC reaction of the dinuclear **Ac2** with **5** was studied through the $\omega$B97XD functional together with the 6-311G(d,p) basis and the LANL2DZ type basis set for Ag and the results give a similar stepwise mechanism to that obtained by using the B3LYP one. This finding confirms the use of B3LYP functional for the study of these AgAAC reactions. Moreover, a real system using phenylacetylene and phenyl azide was investigated in other to validate the reduced model used in this MEDT study. The AgAAC reaction of the dinuclear **Ac4** with **6** was studied and the activation energy associated to the reaction of **Ac4** with **6** via **TS4-1** is found to be 5.7 kcal/mol higher in energy than that associated to the reaction of **Ac2** with **5**. DFT calculations indicate that the 32CA reaction between silver acetylides and azides, following the proposed stepwise path, is an easy reaction in terms of energetics, providing new insights into the mechanistic understanding of the AgAAC reactions. Interestingly, the ionic nature of the starting complexes is herein revealed for the first time, ruling out any covalent interaction involving the silver(I) species throughout the reaction. This finding is supported by the ELF topological analysis of the electronic structure of the stationary points, which also reaffirms the *zw*-type mechanism of the AgAAC reactions. The present combined experimental and theoretical study provides a reciprocal quantitative and qualitative support for both experimental outcomes and MEDT.

**Supplementary Materials:** The following are available online at http://www.mdpi.com/2073-4344/10/9/956/s1, Details and characterization of all prepared 1,2,3-triazole derivatives reported in the text. Figure S1: Energy profile for the uncatalyzed azide-alkyne cycloaddition reaction between methyl alkyne and methyl azide.

**Author Contributions:** Concept and strategy implementation were done by S.-E.S. and H.B.E.A.; synthetic work was performed by L.B., H.B.E.A., and I.F.; computational work was executed by L.R.D., M.R.-G., L.B., and H.B.E.A.; data analysis and interpretation of the catalytic and computational results were done by L.B., H.B.E.A., L.R.D., M.J., and S.-E.S.; writing of the manuscript was carried out by H.B.E.A., L.B., L.R.D., M.J., and S.-E.S. All authors have read and agreed to the published version of the manuscript.

**Funding:** This research received no external funding.

**Acknowledgments:** Financial support from the Spanish Ministerio de Ciencia e Innovación (MCINN) (Projects CTQ2016-75068P) is gratefully acknowledged. M.R.-G. thanks MCINN for a post-doctoral contract co-financed by the European Social Fund (BES-2014-068258).

**Conflicts of Interest:** The authors declare no conflict of interest.

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
