# Peer review of "Deciphering the Mechanism of Silver Catalysis of “Click” Chemistry in Water by Combining Experimental and MEDT Studies†"

_catalysts, doi:10.3390/catal10090956_

Round 1
Reviewer 1 Report
My comments can be found in the attached file.

Author Response
Point-by-point responses to the reviewers:
Reviewer 1:
This manuscript catalysts‐892147 actually is the second resubmission of manuscript catalysts850117 and I will regard it as such. The authors address the mechanism of an interesting reaction, the azide + alkyne 1,3‐dipolar cycloaddition catalyzed by AgCl (AgAAC) in water. They carried out several reactions providing a reasonable scope for future application and computational investigation. The latter was carried out by KS DFT calculations, analysis of DFT reactivity indices and the ELF. My recommendation is that the paper can be published in Catalysts provided that the comments and issues below are taken into account. l. 30: replace click with “click” reaction l.
1. The expression “click” was replaced by “formation” instead of “click reaction” because the last expression is used to describe a reaction between substrates under click chemistry regime, while the former word “click” is a result of the clickable chemistry cycloaddition process.
31: replace ambient conditions with room temperature and pressure
R. The sentence “ambient conditions” was replaced by “room temperature and pressure”.
Lines 33‐34. Description of calculation is not clear. It must be separately clarified (i) whether an ECP for Ag has been used (if so, which ECP? Probably LANL2...) and (ii) which basis set was used for Ag. As it is now described, it seems that no ECP was used and a full electron calculation was carried out with B3LYP.
R. Following the reviewer’s suggestion, the description of calculation has been rewritten as follows:
The Molecular Electron Density Theory (MEDT) was performed by using Density Functional Theory (DFT) calculations at both B3LYP/6-31G(d,p) (LANL2DZ for Ag) and ωB97XD/6-311G(d,p) (LANL2DZ for Ag) levels.
Line 134: the Ks of AgCl is 1.77 10‐10, NOT 1.7 10‐7Lines 136‐137. The sentence “It seems... AAC reaction” must be deleted. There is no evidence that the catalytic species is Ag(I) in solution. Instead, the amount of AgCl used in the reactions (10% mol) is about six orders of magnitude larger than the amount required to saturate 2 ml of water. This strongly suggests that the catalytic species is not Ag(I) in solution. Speculatively, the catalytic species can be suggested to be the surface of AgCl.
R. We thank the reviewer for his/her suggestion, the Ks of AgCl was corrected and now appears as: Ks = 1.77 10‐10.
As I already remarked, the terms Figure and Scheme are used in a confusing way. Please use Scheme whenever depicting a reaction (experimentally carried out). Everything else is a Figure, even when it contains a set of molecular structures. Schemes 3, 4, and 5 actually are Figures and must be renamed. Figure 1 actually is a Scheme and should be renamed accordingly. Scheme 4. This depicts a calculated reaction, not a reaction carried out in the lab so anything above and below the arrow must be deleted.
R. The terms Figure and Scheme are now presented in the revised manuscript as suggested by the reviewer.
Appropriate formal charges must be added to the Lewis structure of methylazide, it is a 1,3‐dipolar species.
R. The formal charges were added to the Lewis structure of methylazide (see Figure 3).
174: see comment to lines 33‐34 Scheme 5. Add the appropriate molecular charge to Ac2, which is a cation.
R. The appropriate molecular charge was added in the structure of Ac2 (see Figure 6).
Lines 273‐275. The activation energy of the reaction involving Ac1 is larger than that of the uncatalyzed reaction. This means that the computational method cannot describe the catalysis or Ac1 is not part of the catalytic cycle. The authors should comment on this point.
R. Following the reviewer’s suggestion, a comment on this point was added as follows: This activation energy of 1.5 kcal/mol found to be above the one associated with the non-catalyzed process (see Supplementary Materials for details), led us to formulate the hypothesis of the dinuclear mechanism pathway.
302: energy profile are in Figure 6, not in Scheme 7.
R. Scheme 7 appears now as a Figure, namely Figure 8 that indicates the energy profile.
Table 3. The graphical part of Table 3 must be moved into a new Figure. Captions have to be changed and created accordingly.
R. Following the reviewer’s suggestion, the graphical part of Table 3 was moved to a new Figure, namely Figure 13.
Reviewer 2 Report
Most of my concerns were adequately addressed by the authors in the revised manuscript. Therefore, I strongly recommend the publication of this manuscript in Catalyst.
Author Response
Reviewer 2
Comments and Suggestions for Authors
Most of my concerns were adequately addressed by the authors in the revised manuscript. Therefore, I strongly recommend the publication of this manuscript in Catalyst.
R. We thank reviewer 2 for his/his recommendation accepting this manuscript for publication as it is.
This manuscript is a resubmission of an earlier submission. The following is a list of the peer review reports and author responses from that submission.
Round 1
Reviewer 1 Report
In this manuscript the authors address the mechanism of an interesting reaction, the azide + alkyne 1,3-dipolar cycloaddition catalyzed by AgCl (AgAAC) in water. They carried out several reactions providing a reasonable scope for future application and computational investigation. The latter was carried out by KS DFT calculations, analysis of DFT reactivity indices and the ELF.
The quality of the manuscript is heterogeneous. Some sections are good as they are and I have very few comments on these (section 1, subsections 2.1 and 2.3, section 3). Other sections are in my opinion questionable because of concerns about the model molecules, the computational methodology, the conclusions, and the textual organization (the whole subsection 2.2 and section 4 = Conclusions). Unfortunately, subsection 2.2 represents the core of the computational section reporting on DFT calculations and reactivity indices. The analysis in 2.3 is based on the electron densities computed in 2.2.
Based on the comments below, my recommendation is that the paper cannot be published in the present form since the computational methodology is in most part not adequate for the aim of investigating a reaction mechanism. Major changes are required, eg, (i) the re-organization of the text so that it follows the data-discussion-conclusion order; (ii) clear, detailed statement of the many limitations of the computational approach employed, it the authors are not willing to switch to a better level of theory and more appropriate model molecules.
General comments
- In general, it seems that the authors started from two pre-set reaction mechanism (Scheme 1), proposed on the basis of literature data, and carried out calculations to support (one of) them. The investigation does sot seem to gave been open to alternative TSs, intermediates, etc.
- The B3LYP/6-31G(d,p)+LALN2DZ(Ag) level of theory to investigate TSs has found wide application in the past but recent comparative studies (eg Mardirossian & Head-Gordon (2017), Molecular Physics, 115:19, 2315-2372) have shown that B3LYP barriers are accurate at +/- 5 kcal/mol. Most people recognize that B3LYP/6-31G(d,p) benefits from error cancellation since functional and basis sets err in opposite directions. However, this is not likely to be as effective in systems including Ag with LANL2DZ as in molecules with atoms up to F or Cl at most. Several recent functionals perform much better (wB97XD, M06-2X, M08-HX,…). Please switch to a better functional or clearly state limitations of B3LYP.
- Since the barrier differences amount to a few kcal/mol, other effects should be included in the computational methodology. The effect of BSSE has not been considered but it can be significant with as small a basis set as 6-31G(d,p). All-electron calculations were performed that neglect relativistic corrections. The latter can be significant for Ag. Use of a relativistic ECP is recommended. Please include corrections in the calculations or clearly state limitations.
- The model molecules of the computational investigation are too small, also in view of the fact that most reactants used are arylazides and arylalkynes. Use of phenylazide and phenylacetylene as model molecules would be much better and well within the computational power of small computers. Eg, the global DFT indices (mu, eta, w, and N) of MeN3 and MeCCH are very different from those of PhN3 and PhCCH. Below it is pointed out how this affects the description of electron flow. Again, please switch to better models or clearly state limitations of the small models used.
- Only electronic-energy barriers DE are shown and discussed. No free energy DG not enthalpy DH barriers are shown despite they have been calculated along with the harmonic analysis. In principle I am not against such a choice but the authors should explain in a detailed way why they chose to discuss only the electronic energy neglecting thermochemical quantities.
- The terms Figure and Scheme are used in a confusing way. Please use Scheme whenever depicting a reaction (experimentally carried out). Everything else is a Figure, even when it just contains a set of molecular structures.
- When reporting on calculations, it is recommended that first the optimized molecular structures are discussed, next their energetics, and finally data are discussed and conclusions drawn. The organization of subsections 2.1 to 2.3 is just reversed.
- English needs improvement. Please work through the text carefully from this perspective.
Sub-section 2.1
The authors carried out several AgAAC providing a reasonable scope. Only 4-substituted triazoles were obtained. The reaction conditions are mild, the workup is simple, and the yields are good.
- I have only a major concern here. Why reactions were not carried out *without* AgCl? Catalyzed reactions must be compared with uncatalyzed reactions. Besides, 1,3-dipolar cycloadditions, including AAC, are strongly accelerated in water probably because reactants are squeezed together by the hydrophobic effect (they form mixed droplets being not soluble in water). I think it is necessary to carry out at least the benzylazide + phenylacetylene reaction in water/RT/24h without AgCl.
Minor issues.
- Nowhere is explicitly stated that 5-substituted triazole were not obtained.
- Figure 1 (should be a Scheme). Use exponents for generic substituent R1, R2. Check the labels of product 3. Insert textual list of substituents, ie, R1 =Bn, 4-Me-Bn, etc
Sub-section 2.2.1
- The acetylide species Ac1/2/3 seems to be pre-set before the actual calculations. Were other acetylide searched?
- Furthermore, Ac1/2/3 seem suitable acetylides for a homogeneous reaction in water. In the present case neither reactant not catalyst is soluble in water. Did the authors considered that the reaction may occur on the surface of solid AgCl? In this case, the simplest model would be species such as MeCC-AgCl and the like. It is also puzzling that the structures in Scheme 3 differ from that in Scheme 5.
- As mentioned above, MeN3 and MeCCH are not good models for the experimental reactions. The CDFT indices are very different from those of PhN3 and PhCCH. The table below shows that the differences of mu, ecc between the X-N3 and X-CCH are large when X=Me but very small when X=Ph. The reaction between PhN3 and PhCCH is apolar. So, the data in subsection 2.2.1 do not describe accurately the experiment.
Dmu Deta Dw DN
Me -1.14 -2.53 0.77 0.12
Ph -0.07 -0.32 0.12 0.09
- Ac2 is a cationic species being formally formed from a MeCC- carbanion, two Ag+ cations and two water molecules. How were the Parr functions calculated? Using the neutral and dicationic species?
- It is very surprising that the Parr functions of cationic Ac2 and neutral Ac3 are so similar. The authors should check and discuss this issue in the text.
- It seems that the conclusion of this long discussion is that the model reactions are at most slightly polar. Then, why use a CDFT methodology developed for polar reactions? Other CDFT methodologies are available. Please consider changing methodology or clearly stress the limitation of the adopted approach.
- Lines 152-154 This sentence is not clear, what does it mean?
Sub-sections 2.2.2 and 2.2.3
The above general comments about the level of theory, the model molecules, thermochemistry, and the textual organizations apply here.
- The activation energy of the reaction involving Ac1 is larger than that of the uncatalyzed reaction. This means that Ac1 is *not* a catalyst. The authors should comment on this point.
- Do TS1 and TS1r directly lead to the products AT1 and AT1r? This should be checked by IRC calculations.
- For Ac2 and Ac3, only the reactive channel leading to the 4-substituted isomer is calculated. Given that Ac1 was earlier shown not to be a catalyst, the regioselectivity of the catalyzed reaction can only be investigated by calculating both regioisomeric TSs involving Ac2 and Ac3.
- As mentioned before, it is surprising that the reactions of 5 with cationic Ac2 and neutral Ac3 have very similar energy profiles. A detailed check and comment are in order here.
- Scheme 6, The broken bonds should be solid bonds.
- Figure 6, Check the labels, CI2 -> IC2 etc
Section 3.
- The ELF analysis is competently carried out and interpreted. It seems that ELF was calculated from the electron densities in the gas phase. Why not use the SCRF=Water densities?
- The authors stress that Ac3 actually is an ion pair, comprising cation AgH2O+ and carbanion MeCC–. I think that this bonding situation cannot exist in water: the carbanion would very quickly react with water. So, this result cannot be transferred to the real systems and reactions experimentally investigated.
Conclusions.
The concern with this section is that the conclusions are in part based on computational data that do not mirror the experimental reactions.
- In several places, conclusion are given that are not supported by the data obtained in this paper. For instance, what supports that “The experimental results are correctly explained by means of the more favourable interactions taking place along the 1,4 reaction path”? The reaction involving Ac1 is not catalyzed and for those involving Ac2 and Ac3 *only* the 1,4 reaction path has been considered.
- “the ionic nature of the starting complexes is herein revealed for the first time, ruling out any covalent interaction involving the silver(I) species throughout the reaction”. What supports this claim? Do the authors mean that a carbanion is stable in water?
- Sometime misleading sentences are present. “the non-catalysed 32CA reaction between 4 and 5 performed in water presents a high activation energy and a poor regioselectivity, yielding 1,5-disubstituted-1,2,3-triazolide” sounds like the authors are escribing experimental results. It should be made very clear what is experimental and what is computational.
- Note that catalysis amount to a lowering of the barrier by 2 kcal/mol, well below the accuracy of B3LYP.
Author Response
Point-by-point responses to the reviewer 1:
In this manuscript the authors address the mechanism of an interesting reaction, the azide + alkyne 1,3-dipolar cycloaddition catalyzed by AgCl (AgAAC) in water. They carried out several reactions providing a reasonable scope for future application and computational investigation. The latter was carried out by KS DFT calculations, analysis of DFT reactivity indices and the ELF.
The quality of the manuscript is heterogeneous. Some sections are good as they are and I have very few comments on these (section 1, subsections 2.1 and 2.3, section 3). Other sections are in my opinion questionable because of concerns about the model molecules, the computational methodology, the conclusions, and the textual organization (the whole subsection 2.2 and section 4 = Conclusions). Unfortunately, subsection 2.2 represents the core of the computational section reporting on DFT calculations and reactivity indices. The analysis in 2.3 is based on the electron densities computed in 2.2.
Based on the comments below, my recommendation is that the paper cannot be published in the present form since the computational methodology is in most part not adequate for the aim of investigating a reaction mechanism. Major changes are required, eg, (i) the re-organization of the text so that it follows the data-discussion-conclusion order; (ii) clear, detailed statement of the many limitations of the computational approach employed, it the authors are not willing to switch to a better level of theory and more appropriate model molecules.
General comments
1. In general, it seems that the authors started from two pre-set reaction mechanism (Scheme 1), proposed on the basis of literature data, and carried out calculations to support (one of) them. The investigation does not seem to have been opened to alternative TSs, intermediates, etc.
Authors: Based on our previous reports and the literature data the two mechanism shown in Figure 1 are the most favourable for [3+2] cycloaddition reaction of azide-alkyne catalyzed by silver.
2. The B3LYP/6-31G(d,p)+LALN2DZ(Ag) level of theory to investigate TSs has found wide application in the past but recent comparative studies (eg Mardirossian & Head-Gordon (2017), Molecular Physics, 115:19, 2315-2372) have shown that B3LYP barriers are accurate at +/- 5 kcal/mol. Most people recognize that B3LYP/6-31G(d,p) benefits from error cancellation since functional and basis sets error in opposite directions. However, this is not likely to be as effective in systems including Ag with LANL2DZ as in molecules with atoms up to F or Cl at most. Several recent functionals perform much better (wB97XD, M06-2X, M08-HX,…). Please switch to a better functional or clearly state limitations of B3LYP.
Authors: DFT analysis using B3LYP as a functional leads to similar results like those using Ag and reported in ref. 41 (Organometallics 35, 2589, 2016) for the electronic nature of the mechanism and energies of intermediates as well as transition states. However, we found that while B3LYP is effective to address the stepwise nature of the CuAAC mechanism; other functionals (wB97XD, LCwPBE, M06-2X and M06-L) perform better the concerted mechanism outcome.
3. Since the barrier differences amount to a few kcal/mol, other effects should be included in the computational methodology. The effect of BSSE has not been considered but it can be significant with as small a basis set as 6-31G(d,p). All-electron calculations were performed that neglect relativistic corrections. The latter can be significant for Ag. Use of a relativistic ECP is recommended. Please include corrections in the calculations or clearly state limitations.
Authors: We agree with the reviewer’s suggestions, but the calculations were performed following recent studies of Cu+ and Ag+ catalyzed 32CA reactions of azides with alkynes such as in the ref.41 (Organometallics 35, 2589, 2016) and the recent (Catalysts 2019, 9, 687).
4. The model molecules of the computational investigatioand Catalysts 2019, 9, 687n are too small, also in view of the fact that most reactants used are arylazides and arylalkynes. Use of phenylazide and phenylacetylene as model molecules would be much better and well within the computational power of small computers. Eg, the global DFT indices (mu, eta, w, and N) of MeN3 and MeCCH are very different from those of PhN3 and PhCCH. Below it is pointed out how this affects the description of electron flow. Again, please switch to better models or clearly state limitations of the small models used.
Authors: As, we replied in point 2. DFT analysis using B3LYP functional for the AgAAC works fine for the model cycloaddition reaction between MeN3 and MeCCH.
5. Only electronic-energy barriers DE are shown and discussed. No free energy DG not enthalpy DH barriers are shown despite they have been calculated along with the harmonic analysis. In principle I am not against such a choice but the authors should explain in a detailed way why they chose to discuss only the electronic energy neglecting thermochemical quantities.
Authors: Many recent theoretical studies have shown, from one hand, that relative enthalpies are closer to relative energies in solvent. On the other hand, free energies demand the exact knowledge of the aggregation of species in solution as this may influence the calculations of entropies.
6. The terms Figure and Scheme are used in a confusing way. Please use Scheme whenever depicting a reaction (experimentally carried out). Everything else is a Figure, even when it just contains a set of molecular structures.
Authors: Following the reviewer’s recommendation, the use of Figure and Scheme was revised.
7. When reporting on calculations, it is recommended that first the optimized molecular structures are discussed, next their energetics, and finally data are discussed and conclusions drawn. The organization of subsections 2.1 to 2.3 is just reversed.
Authors: The subsections 2.1 and 2.3 were organized as suggested by the reviewer.
8. English needs improvement. Please work through the text carefully from this perspective.
Authors: We proceeded checking both spelling and style of the whole manuscript.
Sub-section 2.1
The authors carried out several AgAAC providing a reasonable scope. Only 4-substituted triazoles were obtained. The reaction conditions are mild, the workup is simple, and the yields are good.
1. I have only a major concern here. Why reactions were not carried out *without* AgCl? Catalyzed reactions must be compared with uncatalyzed reactions. Besides, 1,3-dipolar cycloadditions, including AAC, are strongly accelerated in water probably because reactants are squeezed together by the hydrophobic effect (they form mixed droplets being not soluble in water). I think it is necessary to carry out at least the benzylazide + phenylacetylene reaction in water/RT/24h without AgCl.
Authors: The control experiment in the cycloaddition of benzyl azide (1a) and phenylacetylene (2a) in the lack of the silver(I) salt (using water as reaction solvent at room temperature) was investigated and no 1,2,3-triazole was obtained (please, see page 3: sub-section 2.1)
Minor issues.
2. Nowhere is explicitly stated that 5-substituted triazole were not obtained.
Authors: This experimental observation is now added on page 3, sub-section 2.1.
3. Figure 1 (should be a Scheme). Use exponents for generic substituent R1, R2. Check the labels of product 3. Insert textual list of substituents, ie, R1 =Bn, 4-Me-Bn, etc
Authors: The labels of the product 3 were added in Scheme 2.
Sub-section 2.2.1
1. The acetylide species Ac1/2/3 seems to be pre-set before the actual calculations. Were other acetylide searched?
Authors: Acetylide species Ac1/2/3 were obtained after full optimizations of different combination of positions of Ag+, H2O and Cl- with respect to CH3CC-. No other stable structures were found.
2. Furthermore, Ac1/2/3 seem suitable acetylides for a homogeneous reaction in water. In the present case neither reactant not catalyst is soluble in water. Did the authors considered that the reaction may occur on the surface of solid AgCl? In this case, the simplest model would be species such as MeCC-AgCl and the like. It is also puzzling that the structures in Scheme 3 differ from that in Scheme 5.
Authors: As suggested by the reviewer, the Scheme 3 was corrected by the modification of solid bonds by broken ones.
3. As mentioned above, MeN3 and MeCCH are not good models for the experimental reactions. The CDFT indices are very different from those of PhN3 and PhCCH. The table below shows that the differences of mu, ecc between the X-N3 and X-CCH are large when X=Me but very small when X=Ph. The reaction between PhN3 and PhCCH is apolar. So, the data in subsection 2.2.1 do not describe accurately the experiment.
Dmu Deta Dw DN
Me -1.14 -2.53 0.77 0.12
Ph -0.07 -0.32 0.12 0.09
Authors: The comparison done by the reviewer shows that methyl azide and propyne are more reactive than phenyl azide and phenylacetylene, because a high difference in the electrophilicity or nucleophilicity for the reagents involved in the reaction indicates the better reactivity and vice versa.
4. Ac2 is a cationic species being formally formed from a MeCC- carbanion, two Ag+ cations and two water molecules. How were the Parr functions calculated? Using the neutral and dicationic species?
Authors: Indeed, the neutral and dicationic species were used to compute the Parr functions.
5. It is very surprising that the Parr functions of cationic Ac2 and neutral Ac3 are so similar. The authors should check and discuss this issue in the text.
Authors: In these reactions, Ac2 and neutral Ac3 act as nucleophilic species. As shown in Figure 2, the nucleophilic Parr functions (in blue) of Ac2 and Ac3 are not that different.
6. It seems that the conclusion of this long discussion is that the model reactions are at most slightly polar. Then, why use a CDFT methodology developed for polar reactions? Other CDFT methodologies are available. Please consider changing methodology or clearly stress the limitation of the adopted approach.
Authors: The analysis of the polar character of a 32CA reaction is given at the TSs. Although RCs present an unappreciable GEDT, the TSs have polar character; 0.28 e at RC1 and TS1 and 0.16, 0.18e at TS2-1, TS3-1, respectively.
7. Lines 152-154 This sentence is not clear, what does it mean?
Authors: This sentence has been rewritten as follows: Then, the formation of a van der Waals complex before the oxidative coupling step of the metal-acetylide and azide has been questioned in several Ru-, Cu- and Zn-mediated cycloaddition reactions. In fact, its presence in the cycloaddition reaction path was found to depend on the metal, ligands and reaction parameters.[38–41]
Sub-sections 2.2.2 and 2.2.3
The above general comments about the level of theory, the model molecules, thermochemistry, and the textual organizations apply here.
1. The activation energy of the reaction involving Ac1 is larger than that of the uncatalyzed reaction. This means that Ac1 is *not* a catalyst. The authors should comment on this point.
Authors: A comment on this remark was done and now appears as: ‘’The activation energy of the mononuclear path is higher than the uncatalyzed one, which proves that the dinuclear path is most favourable than the mononuclear one’’. ( page 18 line 539-541)
2. Do TS1 and TS1r directly lead to the products AT1 and AT1r? This should be checked by IRC calculations.
Authors: Yes, TS1 and TS1r directly lead to the products AT1 and AT1r; after passing these TSs, the approach of the N3 nitrogen to the C5 carbon atom yields directly the formation of AT1 and AT1r
3. For Ac2 and Ac3, only the reactive channel leading to the 4-substituted isomer is calculated. Given that Ac1 was earlier shown not to be a catalyst, the regioselectivity of the catalyzed reaction can only be investigated by calculating both regioisomeric TSs involving Ac2 and Ac3.
Authors: In these Ag-catalyzed 32CA reactions, the formation of Ac2 and Ac3 determines the regioselectivity of these 32CA-type reactions. Although other the Acs are feasible (see the mononuclear mechanism), the total regioselectivity found in the mononuclear mechanism permits to avoid the study of the regioisomeric reaction paths in the dinuclear mechanism.
4. As mentioned before, it is surprising that the reactions of 5 with cationic Ac2 and neutral Ac3 have very similar energy profiles. A detailed check and comment are in order here.
Authors: Yes, there is a very similar energy deference between the reactions of 5 with cationic Ac2 and neutral Ac3. Note that the geometries of the stationary points involved in the two reaction paths are also very similar, indicating that the substitution of a water molecule for a chlorine anion does not modify substantially the catalytic effect in the corresponding Ac2 and Ac3.
5. Scheme 6, The broken bonds should be solid bonds.
Authors: The broken bonds in Scheme 6 now appear as solid ones.
6. Figure 6, Check the labels, CI2 -> IC2 etc
Authors: The labels were checked throughout all the text.
Section 3.
1. The ELF analysis is competently carried out and interpreted. It seems that ELF was calculated from the electron densities in the gas phase. Why not use the SCRF=Water densities?
Authors: Usually, solvent effects modify energies but do not modify substantially the geometries, which depend on of their electronic structure. So, we usually perform the ELF analyses on the gas phase optimized geometries.
2. The authors stress that Ac3 actually is an ion pair, comprising cation AgH2O+ and carbanion MeCC–. I think that this bonding situation cannot exist in water: the carbanion would very quickly react with water. So, this result cannot be transferred to the real systems and reactions experimentally investigated.
Authors: We agree with the reviewer’s comment that carbanions are not stable in water, but propyne is an alkyne relatively acid, and in addition, the presence of a metal salt (precatalyst) can stabilize the formation of Ac3.
Conclusions.
The concern with this section is that the conclusions are in part based on computational data that do not mirror the experimental reactions.
1. In several places, conclusions are given that are not supported by the data obtained in this paper. For instance, what supports that “The experimental results are correctly explained by means of the more favourable interactions taking place along the 1,4 reaction path”? The reaction involving Ac1 is not catalyzed and for those involving Ac2 and Ac3 *only* the 1,4 reaction path has been considered.
Authors: Unlike the 32CA reactions of azides and alkynes, which are poorly regioselective, those catalyzed by Cu or Ag are completely regioselective (see Organometallics 35, 2589, 2016 and Catalysts 2019, 9, 687). This behavior is in complete agreement with complete regioselectivity found the present work. Note that the nucleophilic Parr functions of Ac1 and AC2 are very similar, expecting then a similar regioselectivity outcome.
2. “the ionic nature of the starting complexes is herein revealed for the first time, ruling out any covalent interaction involving the silver(I) species throughout the reaction”. What supports this claim? Do the authors mean that a carbanion is stable in water?
Authors: Carbanions are not stable in water, but their complexes with Ag, namely Ag-acetylide, do so.
3. Sometime misleading sentences are present. “the non-catalysed 32CA reaction between 4 and 5 performed in water presents a high activation energy and a poor regioselectivity, yielding 1,5-disubstituted-1,2,3-triazolide” sounds like the authors are escribing experimental results. It should be made very clear what is experimental and what is computational.
Authors: This sentence was reformulated, in the conclusion section, as follows: At one hand, while the uncatalysed 32CA reaction between 4 and 5 presents a high activation energy and a poor regioselectivity, yielding 1,5-disubstituted-1,2,3-triazolide, the AgAAC reactions are favoured by 2 kcal/mol, yielding 1,4-disubstituted-1,2,3-triazoles in a regioselective manner, as evidenced by the experimental findings.
4. Note that catalysis amount to a lowering of the barrier by 2 kcal/mol, well below the accuracy of B3LYP.
Authors. The energy difference of 2 kcal/mol is between the mononuclear and dinucelar mechanisms, and not with the uncatalysed reaction. We agree that the energy results do not show a clear lowering of the activation energies, but the proposed mechanism account for the complete regioselectivity in the formation of the 1,4-disubstituted-1,2,3-triazoles.
Reviewer 2 Report
See attached file.

Author Response
Point-by-point responses to the reviewer 2:
In this work, the authors studied the alternative mechanistic pathways for the click of 1,2,3-triazole derivatives by Ag(I)-catalyzed azide-alkyne cycloaddition (AgAAC) reaction.
Both theoretical and experimental techniques were used in this study.
The theoretical study was carried out, using the Molecular Electron Density Theory (MEDT), at an appropriate B3LYP/6-31G(d,p) (LANL2DZ for Ag) DFT level. The authors used methylated species (propyne + methyl azide → 1,4-dimethyl-1,2,3-triazole) as molecular models for these studies.
In the experimental study, several 1,4-disubstituted-1,2,3-triazoles were prepared with high yields according to the theoretical predictions. All these products were easily extracted from the reaction mixture. The products were also appropriately characterized using different complementary techniques (melting temperatures determination, 1H and 13C NMR spectroscopy and mass spectrometry).
This is an interesting work, whose conclusions are very well-supported by the results.
Additionally, the experimental and theoretical results allow appropriate comparisons for reciprocal validation. There is very little detail that would prompt criticism. Therefore, I think that this paper can be accepted for publication in Catalyst with only few minor revisions:
(1) The sentence “The followed straightforward synthetic protocol that leads to the regioselective click of 1,4-disubstituted-1,2,3-triazoles under ambient conditions makes use of
AgCl and water as catalyst and water, respectively.”, presented between lines 29 to 32 of page 1, is not clear and should be reformulate.
Authors: Following the reviewer’s suggestion, such a sentence was reformulated and now appears as «Such a synthetic protocol leads to the regioselective click of 1,4-disubstituted-1,2,3-triazoles in presence of AgCl as catalyst and water as reaction solvent under ambient conditions."
(2) The sentence “The ionic character of the starting compounds, namely Ag-acetylide is revealed for the first time, ruling out any type of covalent interaction that involves the silver(I) complexes, along the reaction pathway.”, presented between lines 38 and 40 of page 1, is too long and somewhat complicated to understand. The authors should divide it in two smaller sentences as follows: “The ionic character of the starting compounds, namely Agacetylide, is revealed for the first time. This rules out any type of covalent interaction, involving the silver(I) complexes, along the reaction pathway.”
Authors: The proposed above sentence was included as: "The ionic character of the starting compounds, namely Ag-acetylide, is revealed for the first time. This rules out any type of covalent interaction, involving the silver(I) complexes, along the reaction pathway."
(3) The authors formatted the stoichiometric indices, used in table 1 of page 5, in plane text.
According to IUPAC rules, they should change this format to subscript.
Authors: The stoichiometric indices in Table 1 were corrected according to IUPAC rules
(4) Some acronyms were used in the text, before their meanings were established. These include:
|
Acronym |
Occurrences |
Definition |
Meaning |
|
CDFT |
Pages 6 and 9 |
Page 16 |
Conceptual DFT |
|
zw |
Pages 1 and 6 |
Page 13 |
Zwitterionic |
|
GEDT |
Pages 6, 8, 9, 12, 13, 14 and 15 |
Page 16 |
Global electron density transfer |
The authors should review carefully the manuscript, for fixing other eventual inconsistencies of this type.
Authors: Following the reviewer’s recommendation, the acronyms used in the mean text of this manuscript (ms) were checked and defined as appeared in there first place of the ms.
(5) There are some typos in the text, such as zwittterionic in page 13. The author should use a spell checker to correct these typos.
Authors: The mean text of the ms was checked in terms of spelling and grammar typos.
Round 2
Reviewer 1 Report
I have read the response of the Authors to my comments. Unfortunately, a satisfactory response to most comments was not given. My recommendation is that this manuscript should not be published in Catalysts.
Issues related to language and graphics have been solved and the blank experiment has been carried out. However, most of the comments related to the computational work were not answered in a satisfactory manner.
In many cases the answer is just: “this has previously been done in that way”. Here is a list:
General comments, points 1, 2, 3, 4, 5
Sub-section 2.2.1, points 2, 3, 5
Sub-sections 2.2.2 and 2.2.3, points 2, 3, 4
Section 3, points 1 and 2
Conclusions: points 1 and 4
I will not comment on each unsatisfactory answer. I just mention a few.
GC / 2: “B3LYP is effective to address the stepwise nature of the CuAAC mechanism; other functionals (wB97XD, LCwPBE, M06-2X and M06-L) perform better the concerted mechanism outcome”. What does that mean? Alternative mechanism must be compared using a single functional. Of course, this can (and should) be done using different functionals to ensure the conclusions are well-founded. If several modern, widely tested DFT functionals point to the concerted mechanism and one well-respected but old functional points to the stepwise mechanism, I would at least be cautious in completely confiding in the old functional…
S2.2.2 / 3: “The comparison done by the reviewer shows that methyl azide and propyne are more reactive than phenyl azide and phenylacetylene, because a high difference in the electrophilicity or nucleophilicity for the reagents involved in the reaction indicates the better reactivity and vice versa“. Experiments were done using are arylazides and arylalkynesor even larger molecules. My comparison shows that the computational models are inadequate, as the response implicitly admits pointing out the large difference of reactivity between real and model reactants.
S 3 / 1: “solvent effects modify energies but do not modify substantially the geometries, which depend on of their electronic structure. So, we usually perform the ELF analyses on the gas phase optimized geometries”. They missed the point: it is the electron density that is analyzed by ELF and it does depend on the solvent (otherwise why bother with SCRF?)
Reviewer 2 Report
All my concerns were adequately addressed by the authors in the revised manuscript. Therefore, I strongly recommend the publication of this manuscript in Catalyst.